# Genetic Algorithm-Based Design for Metal-Enhanced Fluorescent Nanostructures

**DOI:** 10.3390/ma12111766

**Published:** 2019-05-31

**Authors:** Dror Fixler, Chen Tzur, Zeev Zalevsky

**Affiliations:** Faculty of Engineering and the Institute of Nanotechnology and Advanced Materials, Bar-Ilan University, Ramat-Gan 5290002, Israel; dror.fixler@biu.ac.il (D.F.); tzuba123@gmail.com (C.T.)

**Keywords:** metal nanostructures, fluorescence, contrast agents, bio-imaging, genetic algorithms

## Abstract

In this paper, we present our optimization tool for fluorophore-conjugated metal nanostructures for the purpose of designing novel contrast agents for multimodal bioimaging. Contrast agents are of great importance to biological imaging. They usually include nanoelements causing a reduction in the need for harmful materials and improvement in the quality of the captured images. Thus, smart design tools that are based on evolutionary algorithms and machine learning definitely provide a technological leap in the fluorescence bioimaging world. This article proposes the usage of properly designed metallic structures that change their fluorescence properties when the dye molecules and the plasmonic nanoparticles interact. The nanostructures design and evaluation processes are based upon genetic algorithms, and they result in an optimal separation distance, orientation angles, and aspect ratio of the metal nanostructure.

## 1. Introduction

Designing complex fluorophore-conjugated metal nanostructure (FMNS) constructs is an important component in the world of bio-imaging [1]. These constructs allow us to use the plasmonic qualities of metal nanostructures along with the qualities of fluorophores (such as radiative fluorescence lifetime (FLT) and fluorescence intensity (FI)), and allow performing multi-modal bio-imaging, such as FLT imaging microscopy (FLIM) [2,3], diffusion reflection (DR) [4,5], and X-ray computerized tomography (CT) scans [6,7,8], using the same contrast agent. The proximity between plasmonic nanostructures and dye molecules creates special nearfield effects that change the fluorescence intensity, lifetime, and quantum yield of the dye molecule [9], called metal-enhanced fluorescence (MEF) and quenching. The radiative FLT and quantum yield (QY) are affected by the distance and orientation with respect to the nanostructure (NS) [10]. Therefore, optimizing the geometry of the FMNS will improve the effectiveness of the MEF mechanism, resulting in better components for the measurements mentioned above. 

Genetic algorithms (GAs) are an optimization method based on a natural selection process that acts like biological evolution [11]. The algorithm is iterative and modifies a population of individual solutions for the optimization problem, allowing the evaluation of a vast number of possibilities. We propose a method to design FMNS constructs using a GA. GAs are being used in other fields of engineering such as radio frequency (RF) antennas, protein structure engineering, solar cells, and more [12,13,14]. Without a computed search tool, such as a GA, it would be extremely hard to examine most of the possibilities for orientation, distance, and NSs shape.

## 2. Genetic Algorithm

A GA is an iterative method for finding a global extremum to a problem [11,15,16]. It originated in the 1970s like a search algorithm and is lately being used as an optimization method in different fields of engineering like RF, computational biology, plasmonics, and more [12,14]. A GA models the natural process of “survival of the fittest,” starting with an initial “population” and using evaluation, death, reproduction, and mutation sub-processes until it converges to the “best valued” population. The (n + 1)th generation is composed of the fitter population members of the nth generation and their “children” (created from combinations of qualities from both parents and a random change that models mutation).

Each generation consists of a population of M members. In nature, a chromosome is a compact package of genes (actually it is a compact state of DNA molecules, containing the genes). An organism in nature has a constant number of chromosomes, typical for its species. In our work, a nanostructure will be modeled by a single array of parameters, analog to a specimen with one chromosome. Each specimen (population member) has a set of parameters that determine its fitness (the parameters are referred to as “genes”). Every member of the population is evaluated according to a value that is set according to the member’s “genes” through a fitness function. After the evaluation process, the top-valued members go through a “reproduction” process, creating the next generation [13], which involves adding random changes in some of the new members (mutations) and creates the next generation. The GA then selects and builds the next generation based on the high ranked members. 

In our realization of the algorithm, specimens with a better fitness value have a greater chance of reproduction, while the diversity of the new population is maintained by the crossover operator (the method of gene exchange between pairs of specimens in the reproduction stage) and the mutation operator, which creates a new specimen by changing an existing one. The mathematical choice of these operators is the heart of the genetic algorithm implementation. There is more than one way of representing the population members and choosing the different GA operators. Yet, two main principles remain in each of GA’s representations: maintaining the population’s diversity in order to keep a wide solution range and promoting “fitter” solutions (specimens). 

As mentioned earlier, the crossover operator has a profound influence over the efficiency of the optimization using a GA since it may create solutions of the search space that are not accessible to either of the two “parents.” This creation of new solutions that allow for a combination of traits from each “parent” further optimizes each new population, making a GA an efficient search algorithm, mainly because of its traits as a cooperative process (the “interaction” between two different specimens) [17]. MATLAB software (1994-2019 The MathWorks, Inc., Natick, MA, USA) offers a solid genetic algorithm realization through its toolboxes.

## 3. Physical Model

The theory for energy transfer from fluorophores to plasmonic nanostructures is moderately complex, and similar equations have been derived from classical and quantum mechanical considerations. We will describe only the final equations. Readers interested in the physical basis of energy transfer from fluorophores to plasmonic nanostructures are referred to the excellent review by Clegg [18].

Mathematically, the link between the geometrical parameters (coded as the genome) and physical properties that indicate the quality of fluorescence are the fluorescence lifetime (FLT), enhancement rate, and quenching rate. The model is linked to the GA through the cost parameter. The cost parameter is calculated in a way that takes into account the fluorescence parameters. 

The optimization process uses calculations of the scattering cross-section for the nanorod and absorption cross-section for each fluorophore that surrounds the rod. In the heart of the optimization process stands the evaluation function, which checks the value of possible solutions. Every solution (e.g., specimen) is composed of a variation of orientation angles and distance between the fluorophore and the gold nanorod (GNR), as shown in Figure 1, where both shapes were approximated as oval to simplify the amount of information used by the metallic nanoparticle boundary element method (MNPBEM) solver throughout the GA’s iterations. This will affect the specimen’s orientation factor κ (also known as the orientation factor):(1)$$\kappa =\left|cos\left(\theta_{T}\right)-3cos\left(\theta_{D}\right)cos\left(\theta_{A}\right)\right|$$
where $\theta_{T}$ is the angle between the fluorophore and the plasmon dipole moments; $\theta_{D}$ is the angle between the line that connects the fluorophore’s and the GNR’s centers and the plasmon dipole; $\theta_{A}$ is the angle between the line that connects the fluorophore’s and the GNR’s centers and the fluorophore dipole [19].

In the calculations mentioned above, we assumed that the distance is independent of the orientation (zero order dependence approximation). 

First, we calculate the quenching and enhancement as if the fluorophore is right above the upper edge of the GNR with all angles equal to 0. Then, we multiply our result with an orientation factor:$\;\kappa^{2}$ for quenching and $\cos{\left(\theta_{D}\right)}^{2}$ factor for enhancement.

Note that the multiplication by the cosine-squared factor is because an antenna or a dipole radiation field is maximal along the direction of the antenna (or dipole) and it is reduced as a cosine of the orientation angle such that the radiation field is zero in perpendicular to the antenna (or dipole). For intensity, the factor is cosine squared. The rate of transfer from fluorophores to plasmonic nanostructures separated by a distance *r* is dependent on *κ* squared. The *κ*-squared term is a factor describing the relative orientation in space of the transition dipoles of the fluorophore and the nanostructures. More about the orientation factor *κ* can be find in Haas et al [19]. There, one may find the full description of the effect of the orientation of the donor and acceptor on the probability of energy transfer involving electronic transitions of mixed polarizations.

The evaluation function computes the radiative enhancement through an expression, which takes into account the quenching and the MEF effects. The quenching effect is the transfer of energy from the fluorophore to the quencher (in this case, the metallic nanostructure’s plasmons), which causes a decrease in fluorescence. In our case, the fluorophores are fixed around the GNR so that the quenching mechanism should describe a static and distant interaction. A good model for that kind of interaction is the use of a fluorescence-resonance energy transfer (FRET), in which the donor is the fluorophore and the acceptor is the GNR [2], where the quenching energy loss rate is described by Equation (2) and the enhancement energy transfer rate is described by Equation (3). The energy transfer is influenced by the spectral overlap between the GNR absorption and the fluorophore’s emission. This overlap is quantified using an overlap integral that takes into account a molar absorption of the GNR (GNR’s cross section normalized by the total energy, concentration, and cuvette size), and the emission line shape of the fluorophore, described by Equation (2):(2)$$QR\;\left[Hz\right]=\frac{8.79\cdot 10^{-5}\cdot Q_{o}\cdot J_{fp}\cdot \kappa^{2}}{1.34^{4}\cdot \tau_{o}}\cdot \frac{1}{{\left(10\cdot r\left[nm\right]\right)}^{6}}\;J_{fp}=\frac{\int_{-\infty }^{\infty }cross\;section\_NR_{absorbance}\cdot lineshape\_fluorophore_{emission}\cdot \lambda {\left[nM\right]}^{4}d\lambda }{\int_{-\infty }^{\infty }cross\;section\_NR_{absorbance}\cdot d\lambda \cdot C_{NR}\left[M\right]\cdot l\left[cm\right]}$$
where *QR* is the quenching rate, $C_{NR}$ is the nanorod (NR) concentration in the solution (assumed to be = 1 nM), and l is the container (cuvette) size, which equals 1 cm. $J_{fp}$ is the overlapping integral that represents a normalized evaluation standard for the spectral overlap between the lineshapes of the GNR’s absorption plasmon resonance ($cross\;section\_NR_{absorbance}$) and the fluorophore’s photon emission ($lineshape\_fluorophore_{emission}$).

Radiative enhancement was considered using an equation that describes the enhancement of the fluorophore near a gold nanorod (modeled by an ellipsoid) [20]. The GNR material, height, and width affect its absorption and scattering spectra and change the overlap integral. We assumed that the GNR was excited by an electromagnetic field that corresponds to its longitudinal plasmon resonance [21]. We multiplied the normalized enhancement expression by 1/τ_o_ to get the enhancement rate (ER):(3)$$ER={\left|\frac{1+\frac{1}{\left(2r+h\right)\cdot \left(r\left(h+r\right)+R^{2}\right)}-I\left(r\left(h+r\right)\right)}{\frac{1}{\left(\varepsilon_{plasmon\;res}-1\right)hR^{2}}+I\left(0\right)}\right|}^{2}\cdot \cos{\left(\theta_{D}\right)}^{2}\cdot \frac{1}{\tau_{o}}\;I\left(r\right)=\int_{r}^{\infty }\frac{2\cdot dx}{\left(x+R^{2}\right){\left(4x+h^{2}\right)}^{1.5}}$$

After calculating the enhancement rate and the quenching rate of a conjugated structure (a “specimen” in a generation of the GA), we examined two types of optimization processes, where a minimum cost led to a “better” specimen. The first was optimizing the differential enhancement factor ${\;\Gamma }_{d}$ so that high $\Gamma_{d}$ is desirable and the GA’s cost parameter ∝1Γd The second process was based on the assumption that when enhancement conditions are met, the FLT will become equal to $\tau_{0}$ (the natural lifetime) such that maximal FLT is an indication of maximal enhancement. This is an indirect optimization since we use FLT as a way to optimize the enhancement effect. According to the latter, the cost ∝ 1τ, with τ being the lifetime.

The differential enhancement factor (DEF) is set by the difference between the enhancement rate and quenching rate, as described in Equation (4):(4)$$DEF\;\left(\Gamma_{d}\right)=enhancement\;rate-quenching\;rate$$

(5)$$TrE=enhancement\;efficiency=\frac{enhancement\;rate}{enhancement\;rate+quenching\;rate}$$

(6)$$TrQ=quenching\;efficiency=\frac{quenching\;rate}{enhancement\;rate+quenching\;rate}$$

Enhancement efficiency (TrE) and quenching efficiency (TrQ) were determined in order to quantify the effects of quenching and metal enhancement (as shown in Equations (5) and (6)).

The FLT is modeled as a double exponential, while the relative effect of the metal interaction lifetime is a function of the efficiencies and the natural lifetime $\tau_{0}$. Our tool evaluated the level of spectral overlap. We experimented on different nanostructures and different dye molecules and examined each of its possible combinations. 

A scheme for our FLT optimization process is presented in Figure 2.

If the level of overlap is larger or equal to 40%, we presumed that metal enhancement is the dominant effect. We used different expressions for Const1(TrE,TrQ) in order to have a function that fits the expected behavior. If the level of overlap was smaller than 40%, we presumed that quenching was the dominant effect and the weight of the quenching related element was a function of the quenching efficiency (TrQ) only. However, note that the base assumption that quenching is dominant due to spectral overlap being low is not accurate.

## 4. Validation Procedure

Each generation’s population can be described as a “gene-pool,” which is an array containing all of the current generation’s “genetic” information. Population size is the size of the gene-pool vector, given by Equation (7):(7)$$Num\;of\;genes\;in\;pop=N_{specimens}\times \underset{genes\;in\;one\;specimen}{\underset{︸}{M}}$$

A data structure representing a fluorophore–GNR (FGNR) structure contains information about the number of fluorophores, their distance from the GNR, and their angular position with respect to the GNR long axis (e.g., one specimen). Each specimen gets evaluated according to its fluorescence level or QY. The specimens that are evaluated with the highest “grades” are selected as parents and their “genetic data” is transferred to their offspring. The procedure repeats itself until it reaches the highest radiative emission coefficient ($\Gamma_{r})$ or maximum number of generations. This process is illustrated by Figure 3.

The GA optimization was done using a MATLAB solver that receives an evaluation (target) function that is created by the user. The structure of the genome, as well as other optimization parameters, are determined by the main script before the optimization function activates the GA solver. The optimization function also calculates the magnitude of spectral overlap, depending on the nanostructure’s shape and material, and the scattering and absorption spectra of the dye molecule. An indication whether the spectral overlap is above the threshold is then passed to the evaluation function for further calculations of fluorescence parameters and for the estimation of the “cost” parameter, which is needed for the evaluation of each specimen (conjugated structure).

The evaluation function is the link between the physical model’s equations and the GA. The physical equations provide the function quantitative data for parameters like enhancement rate, FLT, and Forster distance. After the calculation of several fluorescence parameters, the evaluation function calculates and returns the GA’s cost parameter for a given specimen. The genome of each specimen is transferred to the evaluation function for the calculation of the cost parameter for this specimen, allowing the GA to evaluate its fitness. 

Genes 1–5 are analogous to a genotype, where each of them has a role in expressing the physical separation distance. The separation distance is a geometric average of the genes, which will be used after its calculation in our fluorescence equations. Genes 6–8 contain information regarding the orientation of the fluorophore’s absorption dipole (as seen in Figure 4). The angles obtained from genes 6–8 are used for the calculation of the orientation factor κ. 

The cost parameter is calculated differently according to the fluorescence property we wish to optimize. We tested two variations, optimizing $\Gamma_{d}$, via the cost function of 1Γd, and optimizing FLT, via the cost function of 1τ.

After an optimal conjugated structure is chosen, our tool copies its genetic data and modifies only the genes related to the separation distance in order to view the fluorescence parameters of every possible solution within a given range of distances. 

MNPBEM was used as our spectral analysis tool. We compared our results to spectrums of the same shapes, as calculated in El Sayed’s work [22]. Spectral analysis is needed for the calculation of the spectral overlap ratio and integral, which are important for the optimization process. The comparison between a spectrum of a metal nanostructure and the absorption spectrum of fluorescein can be seen in Figure 5.

## 5. Numerical Results

### 5.1. Differential Enhancement Factor $\left(\Gamma_{d}\right)$ Optimization

GA optimization was done for FGNS structures composed of a 60 nm × 25 nm GNR, labeled using fluorescein. These structures were represented as arrays of decimal values (e.g., “genes”) representing relative angles of orientation between the fluorophore’s and the GNR’s dipoles and the separation distance of the fluorophores from the GNR. 

Assuming that the number of fluorophores affects the fluorescence intensity and not the enhancement rate or quenching rate, we can compute an optimized solution containing a single fluorophore and apply the same solution to a different amount of fluorophore. Our algorithm was set to populations of 300 members, 550 generations, and a mutation probability of 10%. Simplicity of the genetic structure, resulting from using the same GNR and modifying only the fluorophore’s distance and dipole orientation, allowed the use of large populations. This contributed to the important diversity factor of the GA.

We compared our GA-designed structure to similar structures that were different only by one orientation angle $\theta_{T}$, and scanned for fluorescence enhancement values over different separation distances. Figure 6 demonstrates that the FGNS structure that was suggested by the GA that had a far better enhancement rate than similar FGNS structures that were modified only by the angle $\theta_{T}$.

Please note that in the optimizations of Figure 6
$\theta_{D}$ and $\theta_{A}$ were held constant and were not fully optimized, while $\theta_{T}$ was. However, the result we got for $\theta_{T}$ is actually a local extremum and not a global one because it is only a part of the whole genome of the structure. In order to find an optimal genome, further numerical evolution is needed, which means running the genetic optimization with proper computing power and examining much larger span of possibilities. Thus, the result presented in Figure 6, in this respect, is only preliminary proof of capability and not a full optimization process which is numerically much more complicated.

### 5.2. FLT Optimization

In this experiment, we calculated the FLT according to Equation (8). If the spectral overlap percentage was over the threshold (40%):(8)τ=Const1(TrE,TrQ)·τ0+τ0Const1=(TrE·TrQ4)1/3·exp(enhancement rate−τ0enhancement rate+τ0)−TrQ2τ=1(quenching rate+enhancement rate+1τ0)

We assumed that maximum FLT indicates maximum fluorescence enhancement because when the MEF state occurs, the dye molecule functions as an acceptor in a FRET couple, whereas the plasmon modes of the metal transfer energy in a similar way to a donor molecule. This led us to think that the FLT can be longer than $\tau_{0}$ because of new energy decay possibilities.

We used our optimization tool for three types of dye molecules (fluorescein, rhodamine green, rhodamine red) [23] and a few formations of plasmonic nanostructures: Ag sphere-shaped nanosparticle (SNP) with a 10 nm radius; Ag oval shaped nanorod (SNR) with a diameter product of 40 nm × 20 nm; Au sphere shaped nanoparticle (GNP), also with 10 nm radius; and Au nanorod (GNR), also oval shaped with diameter product of 40 nm × 20 nm. The shape and material of the nanostructures affected their plasmonic properties and the spectral overlap as a result. Our tool operated the GA and provided us with an “optimal” genome for each conjugation of dye and nanostructure. For every optimization process, only the distance-related genotype (genes 1–5) was changed through a naive scan, from 1 nm to 100 nm with a resolution of 25 nm, and the rest of the genes remained unchanged [24]. When the spectral overlap ratio was below 40%, it was assumed that TrE = 0 (MEF was neglected) and only quenching was taken into account and vice versa for spectral overlap ration above 40%. 

Our results are presented in Figure 7. In Figure 7, we see the effect of separation distance on the fluorescence properties of fluorescein-conjugated to GNP as seen in Figure 7a–d. The spectral overlap ratio was below 40%, thus we followed Equation (8). We can see that our assumption fit the physical model we used, as the magnitude of the quenching rate in Figure 7c was much larger than the enhancement rate of Figure 7b.

Note that fluorescence quenching refers to any process that decreases the fluorescence intensity of a sample. A variety of molecular interactions can result in quenching. These include excited-state reactions, molecular rearrangements, energy transfer, ground-state complex formation, and collisional quenching. Figure 7a deals with energy transfer only. The FLT arrived at the 4-ns value after all processes had ended. 

One can observe that quenching was weakened for separation distances larger than ≈10 nm such that FLT rose up to $\tau_{0}$. We observed familiar results in other cases of low spectral overlap, in which quenching was presumed to be the dominant plasmon–fluorophore effect, as seen in the cases of fluorescein–GNP (see Figure 7e), fluorescein–GNR (Figure 7f), fluorescein–SNP (Figure 7g), rhodamine G–SNP (Figure 7h), rhodamine R–SNR (Figure 7i), rhodamine R–SNP (Figure 7j), rhodamine R–GNR (Figure 7k) and rhodamine G–GNR (Figure 7l).

As for the cases of conjugation between fluorescein and rhodamine G with a 40 nm × 20 nm silver nanorod (Figure 8 left and right sides), our calculations took MEF into account and we can see that FLT had reached larger values than $\tau_{0}$. According to our model, the point of maximum FLT indicates the optimal separation distance, which was 60 nm for the fluorescein–SNR conjugation and 75 nm for the rhodamine G–SNR conjugation.

Note that such a large distance actually means that there was no coupling. On the other hand, in this manuscript, we present a conceptual approach and not a specific optimal design. Indeed, for the inspected wavelengths and sizes of nanoparticles what we have received means that there was no enhancement and that the optimal separation distance needed to be large. However, for other combinations, such as rhodamine with nanorods, the optimal results produced much closer distances within the range of 8–15 nm.

## 6. Conclusions and Discussions

The evaluation of field intensity over wavelength has shown good correlation with the calculation of lateral intensity dispersion as seen by the brightest intensity graphs matching the peak of the ${\left|E\right|}^{2}$ vs. wavelength graph. We inspected some lateral simulations of ${\left|E\right|}^{2}$. These simulations were performed at a different wavelength in every simulation, some with excitation polarized longitudinally (a–c) and some with transverse polarization (d–f). The highest ${\left|E\right|}^{2}$ values matched the resonance seen in the spectral simulations above. We can observe these results in Figure 9.

The optimization processes we initiated has shown us that our tool provides the expected results optimization-wise and it will surely be beneficial for the future design of contrast agents. However, the quality of its results depends greatly on the validity of the physical model that is being used in its evaluation function. In our FLT optimization, we used FLT as an indicator for optimal MEF interactions and we conclude that a model in which the different energy transfer rates are calculated directly, without the use of circumvention through optimization of FLT, may provide valid results. Nevertheless, our optimization tool has provided a proof of concept for the benefit of machine learning algorithms and may become much more useful with more accurate physical equations. Under the assumptions we made, for a specific FLT and enhancement factor models and assuming that the spectral overlap is the main cause of whether MEF is dominant or quenching is dominant, we could deduce that a silver nanorod would be a good conjugate to a green dye molecule, gold nanostructures would be better for infra-red dye molecules, and silver NPs would work best with purple and UV molecules. 

The physical model that was used does not express the physics accurately enough, thus affecting the quality of our optimization. A more accurate model needs to be introduced in order to allow our optimization tool to provide us with an engineered solution. Choosing FLT as an indicator of maximal enhancement did not provide us with the information we needed. Using the enhancement rate directly will most likely provide better solutions for optimized intensity-based contrast agents. 

As GA optimization effectiveness depends greatly on the structure of the genome, population size, mutation rate, and more, these parameters need to be fine-tuned in order to optimize the process effectively. GAs will most likely provide an acceptable solution, though not necessarily the superior one. GA-based tools, like the one we have created, can be fast, simple, and best suited for multi-variables optimization problems.

## Figures and Tables

**Figure 1 materials-12-01766-f001:**
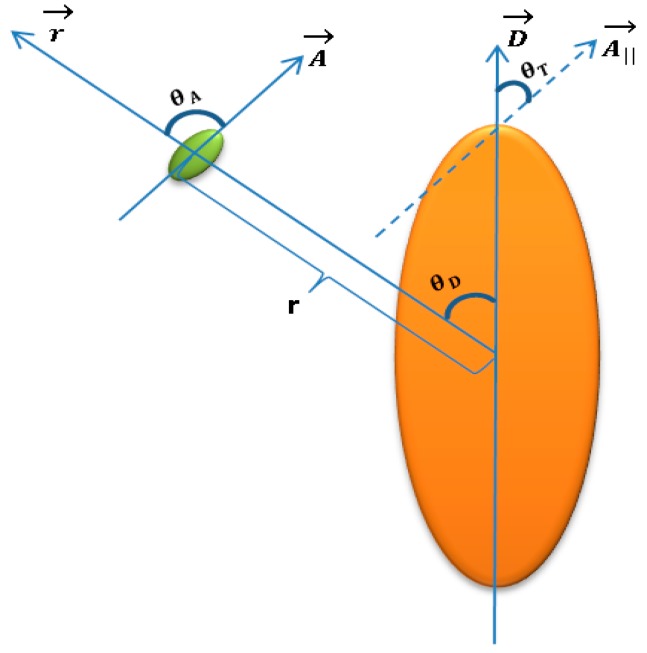
Sketch of fluorophore-GNR structure that shows the parameters that are passed as “genes” to the GA. The fluorescent molecule is in green and the GNR is in orange. Both shapes are approximated as oval to reduce computation complexity.

**Figure 2 materials-12-01766-f002:**
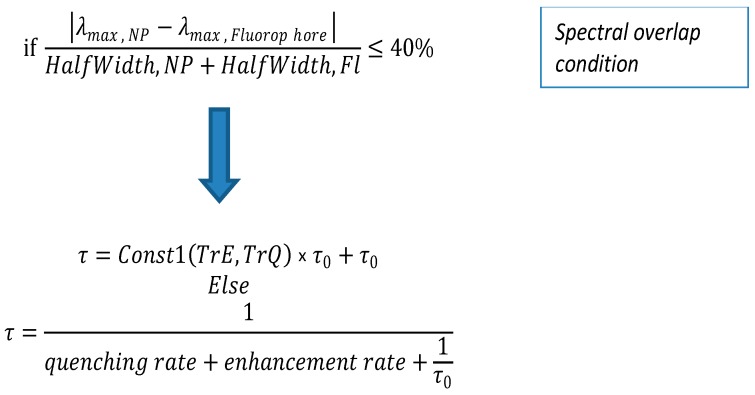
A basic scheme of our physical model.

**Figure 3 materials-12-01766-f003:**
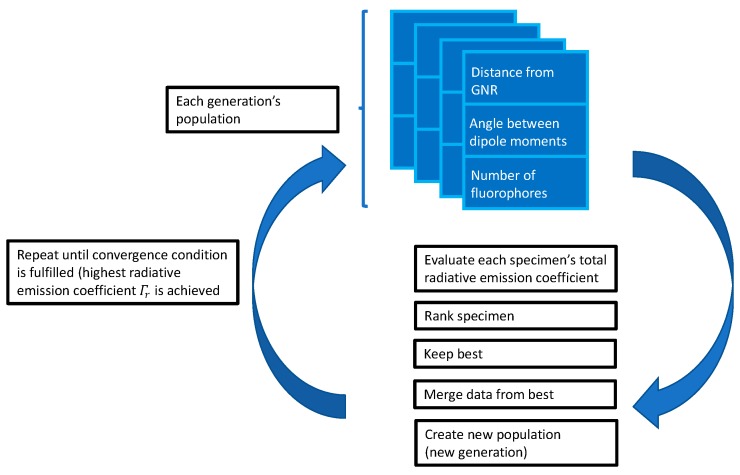
Flowchart of a GA for FGNR optimization.

**Figure 4 materials-12-01766-f004:**
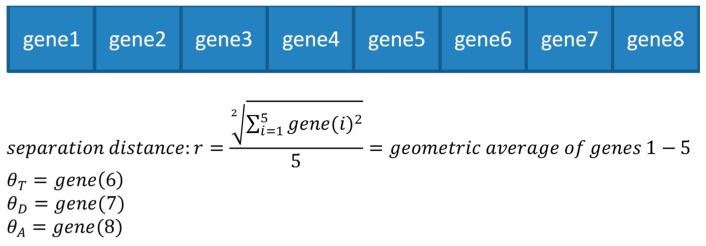
Genome structure and gene expression.

**Figure 5 materials-12-01766-f005:**
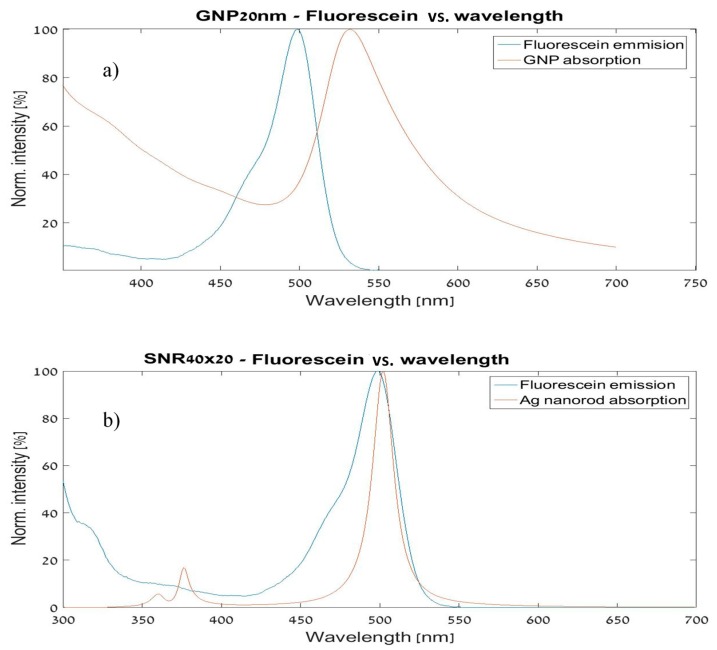
Spectral overlap for fluorescein with (**a**) a GNP and (**b**) a silver nanorod.

**Figure 6 materials-12-01766-f006:**
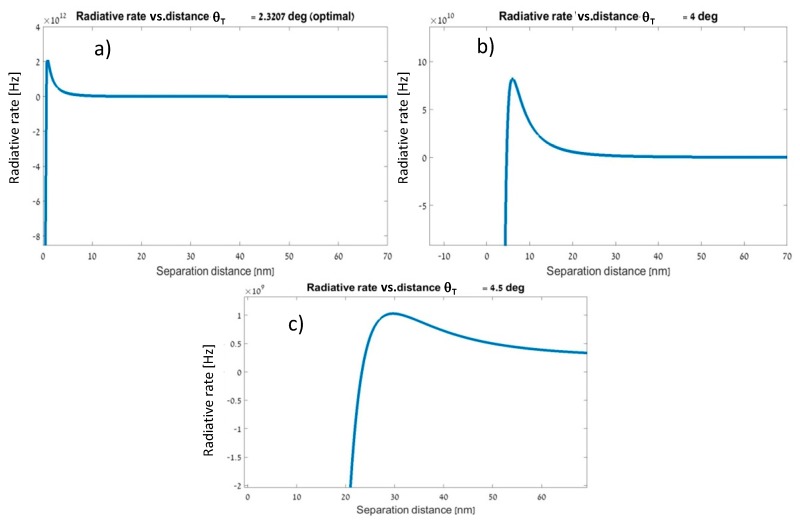
Radiative rate vs. separation distance at different tilt angles ($\theta_{T}$): (**a**) $\theta_{T}=$ 2.3207° (optimal), $\Gamma =2.106\times 10^{12}\;\left[\mathrm{Hz}\right]$, (**b**) $\theta_{T}=$4°, $\Gamma =8.24\times 10^{10}\;\left[\mathrm{Hz}\right]$, (**c**) $\theta_{T}=$4.5°, $\Gamma =1.029\times 10^{9}\;\left[\mathrm{Hz}\right]$. Change in one orientation angle greatly affects the radiative enhancement. A GA-based design shows best results.

**Figure 7 materials-12-01766-f007:**
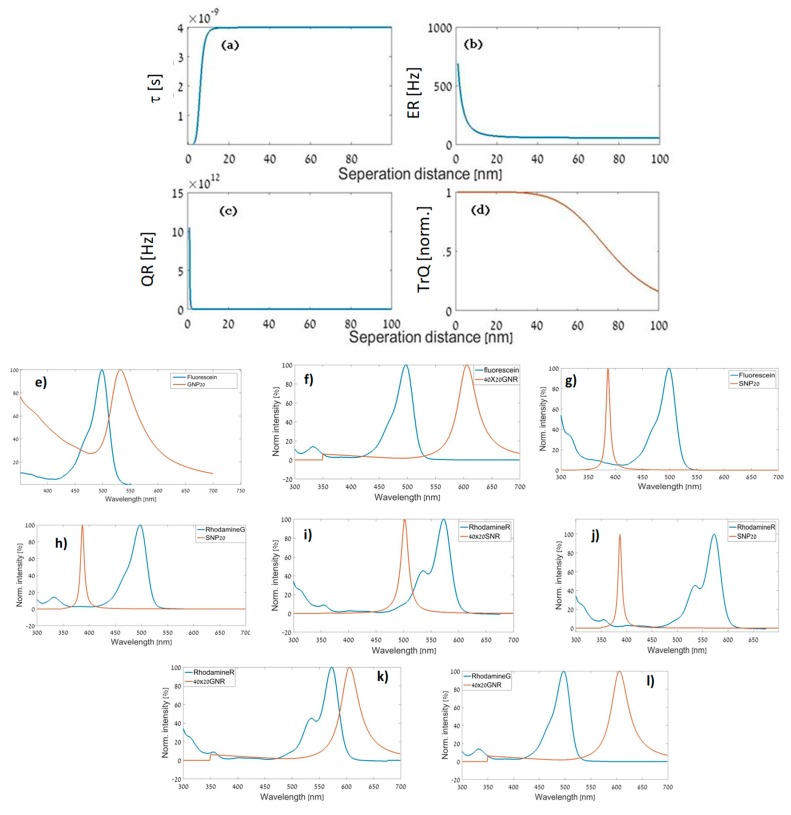
Fluorescence parameters in a state of quenching. (**a**) FLT of the conjugated structure vs. separation distance (spacer thickness) for a fluorescein-GNP structure. (**b**) Enhancement rate vs. spacer thickness for a fluorescein-GNP structure. (**c**) Energy transfer quenching rate vs. spacer thickness for a fluorescein-GNP structure. (**d**) Normalized quenching efficiency vs. separation distance (enhancement is neglected) for a fluorescein–GNP structure. (**e**) Extinction vs. wavelength for a fluorescein–GNP structure. (**f**) Extinction vs. wavelength for a fluorescein–40 nm × 20 nm GNR structure. (**g**) Extinction vs. wavelength for a fluorescein–SNP structure. (**h**) Extinction vs. wavelength for a rhodamine G–SNP structure. (**i**) Extinction vs. wavelength for a rhodamine R—40 nm × 20 nm SNR structure. (**j**) Extinction vs. wavelength for a rhodamine R–SNP structure. (**k**) Extinction vs. wavelength for a rhodamine R—40 nm × 20 nm GNR structure. (**l**) Extinction vs. wavelength for a rhodamine G—40 nm × 20 nm GNR structure.

**Figure 8 materials-12-01766-f008:**
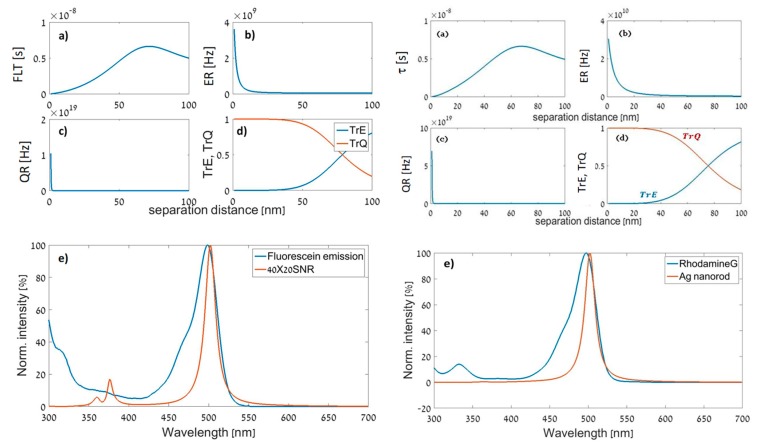
Left: 40 nm × 20 nm Ag NR (SNR) and fluorescein. (**a**) FLT of the conjugated structure vs. separation distance (spacer thickness). (**b**) Enhancement rate vs. spacer thickness. (**c**) Quenching rate vs. spacer thickness. (**d**) Normalized quenching and enhancement efficiencies vs. separation distance. (**e**) Extinction vs. wavelength. Right: Ag NR (SNR) and rhodamine G. (**a**) FLT of the conjugated structure vs. separation distance (spacer thickness). (**b**) Enhancement rate vs. spacer thickness. (**c**) Quenching rate vs. spacer thickness. (**d**) Normalized quenching and enhancement efficiencies vs. separation distance. (**e**) Extinction vs. wavelength.

**Figure 9 materials-12-01766-f009:**
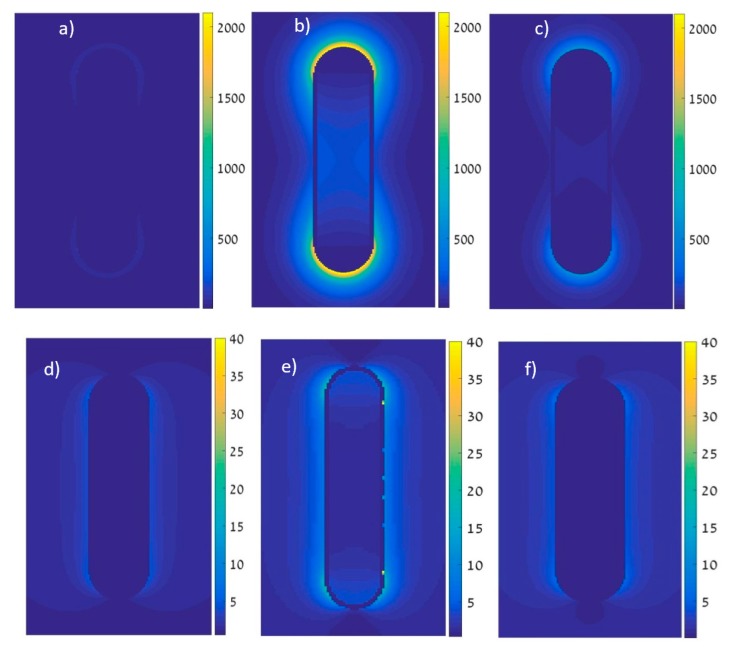
${\left|E\right|}^{2}$ vs. distance from the nanorod. Excitation at (**a**) longitudinal mode, 750 nm; (**b**) longitudinal mode, 840 nm; (**c**) longitudinal mode, 900 nm; (**d**) transverse mode, 400 nm; (**e**) transverse mode, 537 nm (highest intensity); and (**f**) transverse mode, 840 nm.

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
