# Peer review of "Genetic Algorithm-Based Design for Metal-Enhanced Fluorescent Nanostructures"

_materials, 2019, doi:10.3390/ma12111766_

Round 1
Reviewer 1 Report
In this report the authors presented a methodology of optimizing fluorophore-metal nanostructure based on genetic algorithms. Physical models were established to evaluate the effects of the geometric parameters and spectra overlap on the plasmonic enhancement rate, the quenching rate and fluorescence lifetime as well as to optimize the parameter to achieve the best design. The motivation of this work is fascinating, but a few places require clarification before the manuscript becomes suitable for publication.
(1) In figure 1, theta D seems to be the angle between the line connecting centers and the plasmon dipole, and theta A is the angle between the line and fluorophore dipole.
(2) Why the result is multiplied by k2 for quenching and cos(theta D)2 for enhancement (page 2 line 94-95)? This mathematic operation is not straightforward for most of readers and it needs explanations.
(3) What do the constants shown in Eq.2 stand for? The authors should either derive this equation or cite literature. Same issue applies to Eq.3.
(4) The discussion needs to be more self-consistent. In page 3 line 135-138, it is stated that spectra overlap <40% leads to dominant metal enhancement effect while overlap>40% results in dominant quenching effect. However, in page 7 line 225-226, it is assumed that enhancement is negligible when spectra overlap is below 40%.
(5) In figure 6, it would be useful to show results calculated from more theta T, especially when theta T is <2.32 deg. Also, the discussion/optimization of theta D and theta A are missing.
(6) In figure 7a, FLT rises to natural lifetime at distance >10nm, while figure 7c shows that the quenching rate drops to a minimal at distance much smaller than 10nm, how to interpret this discrepancy?
(7) In the first paragraph of conclusion the authors claimed that brightest intensity graphs match E2 vs wavelength graph. I can not find data supporting this conclusion. The authors need to be more specific for this point.
Author Response
In this report the authors presented a methodology of optimizing fluorophore-metal nanostructure based on genetic algorithms. Physical models were established to evaluate the effects of the geometric parameters and spectra overlap on the plasmonic enhancement rate, the quenching rate and fluorescence lifetime as well as to optimize the parameter to achieve the best design. The motivation of this work is fascinating, but a few places require clarification before the manuscript becomes suitable for publication.
We would like to thank the reviewer for his positive evaluation of our work. As can be seen below all his comments were taken into account in the revised manuscript.
(1) In figure 1, theta D seems to be the angle between the line connecting centers and the plasmon dipole, and theta A is the angle between the line and fluorophore dipole.
We agree with the reviewer, we had a mistake in the notation. This mistake was corrected in the revised version of our manuscript.
(2) Why the result is multiplied by k2 for quenching and cos(theta D)2 for enhancement (page 2 line 94-95)? This mathematic operation is not straightforward for most of readers and it needs explanations.
We thank the reviewer for his comment. The multiplication by the cosine square factor is because an antenna or a dipole radiation field is maximal along the direction of the antenna (or dipole) and it is reduced as a cosine of the orientation angle such that the radiation field is zero in perpendicular to the antenna (or dipole). If one computes intensity, then it goes as cosine square. The rate of transfer from fluorophores to plasmonic nanostructures separated by a distance r is dependent of k square. The term k square is a factor describing the relative orientation in space of the transition dipoles of the fluorophore and the nanostructures. More about the orientation factor k can be find at Haas et al (new Ref. 19). There one may find the full description of the effect of the orientation of donor and acceptor on the probability of energy transfer involving electronic transitions of mixed polarizations. Following the reviewer's comment, we have added a new reference as well as a clarification paragraph.
(3) What do the constants shown in Eq.2 stand for? The authors should either derive this equation or cite literature. Same issue applies to Eq.3.
We thank the reviewer for pointing tis out. The constants are related to conversion of units. The theory for energy transfer from fluorophores to plasmonic nanostructures is moderately complex, and similar equations have been derived from classical and quantum mechanical considerations. In our manuscript we described only the final equations. Following the reviewer’s comment we added to our revised manuscript a reference to the excellent review done by Clegg (new Ref. 18)
(4) The discussion needs to be more self-consistent. In page 3 line 135-138, it is stated that spectra overlap <40% leads to dominant metal enhancement effect while overlap>40% results in dominant quenching effect. However, in page 7 line 225-226, it is assumed that enhancement is negligible when spectra overlap is below 40%.
We fully agree with the reviewer. It is a typo mistake which was corrected in the revised manuscript. We assumed that for overlap larger than 40% is metal enhancement and for smaller than 40% quenching is dominant.
(5) In figure 6, it would be useful to show results calculated from more theta T, especially when theta T is <2.32 deg. Also, the discussion/optimization of theta D and theta A are missing.
We thank the reviewer for a good comment. In our simulations Theta_D and Theta_A were held constant and were not fully optimized, while Theta_T was. However, the result we got for Theta_T is actually a local extremum and not a global one because it is only a part of the whole genome of the structure. In order to find an optimal genome, further evolution is needed, which means running the genetic optimization with proper computing power and examining much larger span of possibilities. The result presented in the current manuscript in this respect is only preliminary proof of capability and not a full optimization process which is numerically much more complicated. We intend to further investigate this point but at the limited time scale and computing resources we can not address it in the current revision process. We do intend to further explore this point deeply and make it as a topic for a continuation research work. Proper reservations and modifications were added in respect to this point, in our revised manuscript.
(6) In figure 7a, FLT rises to natural lifetime at distance >10nm, while figure 7c shows that the quenching rate drops to a minimal at distance much smaller than 10nm, how to interpret this discrepancy?
We thank the reviewer for his comment and apologize for the lack of clarity. Fluorescence quenching refers to any process that decreases the fluorescence intensity of a sample. A variety of molecular interactions can result in quenching. These include excited-state reactions, molecular rearrangements, energy transfer, ground-state complex formation, and collisional quenching. Fig. 7(a) deals with energy transfer only. The FLT arrive to the 4ns value after all process ended. Following the reviewer’s comment, we have added a clarification paragraph.
(7) In the first paragraph of conclusion the authors claimed that brightest intensity graphs match E2 vs wavelength graph. I can not find data supporting this conclusion. The authors need to be more specific for this point.
We thank the reviewer and apologize for the luck of support. Proper computation and simulations were added in the revised manuscript.
Reviewer 2 Report
The authors present an original strategy, based on genetic type algorithms, to obtain the correct guidelines for the design and synthesis of dye/metal nanoparticles complexes with enhanced emission properties. Despite the optimization strategy proposed could be in principle interesting, there is a serious lack of results analysis that prevent the publication of this research.
At the end of the paper, the authors says that the best configuration of a dye/silver nanorod complex in order to maximize the radiative lifetime of the emitter is to fix the dye at a distance of 60 nm from the nanorod. Considering that the dye size is about 1-2 nm and they modeled a 40nm x20 nm nanorod, the result of the calculation is meaningless, since it pratically indicates to not couple the two materials.
Similar considerations can be done on the other results presented. Considering that a lot of woirk shoul dbe done on the graph presentation (the for rates is hertz, and the time unit symbol is "s" not "sec"), at the present status, the manuscript cannot be published.
Author Response
The authors present an original strategy, based on genetic type algorithms, to obtain the correct guidelines for the design and synthesis of dye/metal nanoparticles complexes with enhanced emission properties. Despite the optimization strategy proposed could be in principle interesting, there is a serious lack of results analysis that prevent the publication of this research.
We thank the reviewer for considering our approach interesting. We have revised our manuscript and we hope that the reviewer may find the revised manuscript more suitable for publication in its modified form where additional figure and more explanations were added.
At the end of the paper, the authors says that the best configuration of a dye/silver nanorod complex in order to maximize the radiative lifetime of the emitter is to fix the dye at a distance of 60 nm from the nanorod. Considering that the dye size is about 1-2 nm and they modeled a 40nm x20 nm nanorod, the result of the calculation is meaningless, since it practically indicates to not couple the two materials.
We thank the reviewer for a very good comment. In general we agree with the reviewer and indeed such a large distance actually means that there is no coupling. On the other hand, in this manuscript we present a conceptual approach and not a specific optimal design. Indeed, for the inspected wavelengths and sizes of nano particles this is what we received and the meaning is that there is no enhancement and that the optimal separation distance needs to be large. However, for other combinations such as Rhodamine with nano rods, the optimal results produce much closer distances within the range of 8-15nm. Proper clarification was added to the revised manuscript.
Similar considerations can be done on the other results presented. Considering that a lot of work should be done on the graph presentation (the for rates is hertz, and the time unit symbol is "s" not "sec"), at the present status, the manuscript cannot be published.
We thank the reviewer for his comment. The symbols in the various graphs were modified in the revised manuscript.
Reviewer 3 Report
The manuscript entitled “Genetic Algorithms Based Design for Metal-Enhanced Fluorescent Nanostructures” by Fixler et al. demonstrates the genetic algorithm-based proper design of metallic structures that change fluorescence properties when the dye molecules and the plasmonic nanoparticles interact. It is an interesting topic and would be of interest to fluorescence-based technology field. However, obviously, it looks like that the authors have not paid enough attention to the manuscript. Consequently, the manuscript is not in good quality for publication yet. Authors would need to have modified and corrected my concerns before having this manuscript accepted for publication. My concerns are as below
There is lack of information for figure 1. What is orange colored sphere and the green colored sphere? Authors need to clarify those in the figure caption rather than letting readers guess. I guess green colored might be the flurophore and the orange colored object might be the GNR. Also, authors mainly use nanorod for their study, however, the object in figure 1 is sphere shape object. Authors would need to use nanorod structure in their figure 1.
The abbreviation of GNR (on page 2 line 82) for gold nanoparticle is somewhat misleading. GNR sounds like gold nanorod rather than gold nanoparticle. I suggest the authors to use GNP as an abbreviation for gold nanoparticle. Otherwise, if the GNR actually indicate the gold nanorod, I suggest change the “gold nanoparticle” to “gold nanorod”. It provides confusion.
The full name of FRET has not been introduced in the manuscript. Although I believe that most readers in this field would know the full name of FRET, but I think that the full name of FRET should be introduced for consideration of readers who are not familiar with this fluorescence-based technology field.
For equation (2), authors would need to clarify the meaning of the constants used in the equation. What are those constants representing?
On page 3 line 131 & page 5 line 171, it looks like that the references have not been correctly cited. I can find “Error! Reference source not found”
The abbreviation of MNPBEM has been introduced as well without introduction of full name of it before its first use.
On page 7, abbreviations indicating nanoparticles are quite misleading. Silver nanosphere (SNP) would be needed to abbreviate as SNS and AU nanosphere would be GNS rather than GNP. Authors need to pay more attention on abbreviations.
Found another “Error! Reference source not found” on page 7.
Author Response
The manuscript entitled “Genetic Algorithms Based Design for Metal-Enhanced Fluorescent Nanostructures” by Fixler et al. demonstrates the genetic algorithm-based proper design of metallic structures that change fluorescence properties when the dye molecules and the plasmonic nanoparticles interact. It is an interesting topic and would be of interest to fluorescence-based technology field. However, obviously, it looks like that the authors have not paid enough attention to the manuscript. Consequently, the manuscript is not in good quality for publication yet. Authors would need to have modified and corrected my concerns before having this manuscript accepted for publication. My concerns are as below
We thank the reviewer for considering our topic and research interesting and for his time invested to improve the quality and clarity of our manuscript. As can be seen below all the comments of the reviewer were taken into account when revising our manuscript.
There is lack of information for figure 1. What is orange colored sphere and the green colored sphere? Authors need to clarify those in the figure caption rather than letting readers guess. I guess green colored might be the flurophore and the orange colored object might be the GNR. Also, authors mainly use nanorod for their study, however, the object in figure 1 is sphere shape object. Authors would need to use nanorod structure in their figure 1.
We thank the reviewer for his comment. Indeed, in green we mark the fluorescent molecule and in orange, the GNR. Both shapes approximated as oval to reduce computation complexity. Proper clarification was added in the captions of Fig. 1.
The abbreviation of GNR (on page 2 line 82) for gold nanoparticle is somewhat misleading. GNR sounds like gold nanorod rather than gold nanoparticle. I suggest the authors to use GNP as an abbreviation for gold nanoparticle. Otherwise, if the GNR actually indicate the gold nanorod, I suggest change the “gold nanoparticle” to “gold nanorod”. It provides confusion.
We fully agree with the reviewer. Proper corrections were performed in the revised manuscript.
The full name of FRET has not been introduced in the manuscript. Although I believe that most readers in this field would know the full name of FRET, but I think that the full name of FRET should be introduced for consideration of readers who are not familiar with this fluorescence-based technology field.
We fully agree with the reviewer. Proper corrections were performed in the revised manuscript
For equation (2), authors would need to clarify the meaning of the constants used in the equation. What are those constants representing?
We thank the reviewer. The constants in Eq. 2 were better explained in the revised manuscript.
On page 3 line 131 & page 5 line 171, it looks like that the references have not been correctly cited. I can find “Error! Reference source not found”
We thank the reviewer for pointing this out. Those links were deleted in the revised manuscript.
The abbreviation of MNPBEM has been introduced as well without introduction of full name of it before its first use.
The full name was added in the revised manuscript.
On page 7, abbreviations indicating nanoparticles are quite misleading. Silver nanosphere (SNP) would be needed to abbreviate as SNS and AU nanosphere would be GNS rather than GNP. Authors need to pay more attention on abbreviations.
We thank the reviewer. We fixed all of the abbreviations and better explained them.
Found another “Error! Reference source not found” on page 7.
We deleted the link to equation 7 that caused the mentioned error.
Round 2
Reviewer 2 Report
With the changes and clarification made, the manuscript is now a sufficient level to be published.